# Matching-Based Resource Allocation for Satellite–Ground Network

**DOI:** 10.3390/s22218436

**Published:** 2022-11-02

**Authors:** Huixia Ding, Sicheng Zhu, Sachula Meng, Jinxia Han, Heng Liu, Miao Wang, Jiayan Liu, Peng Qin, Xiongwen Zhao

**Affiliations:** 1Information & Communication Department, China Electric Power Research Institute, Beijing 100192, China; 2School of Electrical and Electronic Engineering, North China Electric Power University, Beijing 102206, China

**Keywords:** resource allocation, satellite–ground network, matching theory

## Abstract

With the vigorous development of information and communication technology, mobile internet has undergone tremendous changes. How to achieve global coverage of the network has become the primary problem to be solved. GEO satellites and LEO satellites, as important components of the satellite–ground network, can offer service for hotspots or distant regions where ground-based base stations’ coverage is limited. Therefore, we build a satellite–ground network model, which transforms the satellite–ground network resource allocation problem into a matching issue between GEO satellites, LEO satellites, and users. A GEO satellite provides data backhaul for users, and a LEO satellite provides data transmission services according to users’ requests. It is important to consider the relationships between all entities and establish a distributed scheme, so we propose a three-sided cyclic matching algorithm. It is confirmed by a large number of simulation experiments that the method suggested in this research is better than the conventional algorithm in terms of average delay, satellite revenue, and number of users served.

## 1. Introduction

The next-generation mobile communication network is expected to provide communication services for users with different applications anytime and anywhere, such as smart cities, UAVs, ocean surveillance, Internet of Things, and smart industries [1,2,3]. The satellite–ground network is a promising solution to achieve this goal. Satellite communication can solve the coverage constraints of 5G communication, even in remote areas, it can still be connected, and it has great potential for ultra-broadband connections. It has significant coverage advantages in application scenarios such as oil well platforms, ships, and agriculture [4,5]. Therefore, the use of satellites as transmission nodes in communication networks has become an important part of the next-generation mobile communication system solutions.

Satellites, including geostationary Earth orbit (GEO) and low Earth orbit (LEO), are essential elements in space to deliver services globally. LEO satellites can offer relatively high data rates and quick propagation delays because they are the closest to Earth. However, LEO satellites move quite quickly relative to the Earth, and it is necessary to transfer satellites frequently to maintain coverage. [6]. In contrast, despite the lower data rate and longer propagation delay of the farthest GEO satellites, their geosynchronous motion ensures stable and incredibly wide ground coverage, offering mobile users great mobility assistance [7]. At present, the number of terrestrial terminal users continues to grow rapidly, and the random access of massive terminals can easily lead to network congestion, but the lack of a unified scheduling strategy among satellite networks has brought huge challenges to access control of the users, so how to effectively use the existing resources to meet the increasing user demands has become an opening problem.

By overcoming the limitations of traditional schemes and game theory, matching games have become a potential scheme in network resource allocation [8,9,10,11,12]. There are classical matching schemes between two-party entities [11]. Paper [9] solved the connection problem between electric equipment and UAVs in the power Internet of Things scenario by using a two-sided matching algorithm. It is important to consider the relationships between all entities and establish a distributed scheme. In the satellite–ground network, three types of entities, GEO satellites, LEO satellites, and end users, need to be considered. The three categories of entities must frequently be decoupled by the two-sided matching algorithm, which results in a lack of preference data for different entities and a reduction in system efficiency. There is a three-sided matching scheme between triple entities, and the relationship problem between three entities (GEO satellites, LEO satellites, and end users) can be reformulated using three-sided matching with size and cycle preference list (TMSC) among the three entities matching game [13]. Finding a stable match with the highest cardinality among GEO satellites, LEO satellites, and end users is the objective of TMSC. As it is NP-complete to determine whether a stable match exists in the TMSC model [14], we transformed it into a restricted TMSC by adding some reasonable constraints.

In this paper, we use the three-sided matching theory to solve the satellite–ground network resource allocation problem. Among them, GEO satellites provide users with data backhaul, and LEO satellites provide data transmission services according to users’ requests. Therefore, we propose a three-sided cyclic matching algorithm. The benefits of our suggested solution include the following: (a) in this method, all three entities are considered simultaneously: GEO satellites, LEO satellites, and end users; (b) the distributed nature of matching algorithms makes it possible to effectively alter resource allocation in response to the constantly changing user demands and the time-varying nature of satellites. The main contributions of this paper are as follows:We propose a satellite-oriented resource allocation model, composed of GEO satellites, LEO satellites, and users. Then, we design a TMSC problem and create a matching issue with three sides including GEO satellites, LEO satellites, and users in which each agent creates a list of preferences by sorting those of the other agents;It is NP-complete to determine whether a stable match exists in the TMSC model, and by adding some reasonable constraints, we transform it into a restricted TMSC problem. An algorithm is proposed to solve the above problems. It can find stable results;We conduct a large number of simulation experiments, and compared with the greedy method, the random allocation method, and the two-sided matching method, and verify that the proposed algorithm is effective in delay, satellite revenue, etc.

The rest of this article is organized as follows. Section 2 introduces the related work in recent years. Then, the model of the satellite–ground network resource allocation system is expounded, and in Section 4, maximizing satellite revenue is taken as an optimization problem. In Section 5, a three-sided cyclic matching algorithm is proposed to deal with the optimization problem. Then, in Section 6, the simulation results and corresponding analysis are given. Finally, conclusions are drawn in Section 7.

## 2. Related Work

Recently, to enable seamless mobile communications coverage worldwide, the satellite–ground network attracted tremendous research interest. Reference [15] developed an air–ground integrated network composed of satellites, UAVs, and ground users, which provided users with lower transmission delay. Reference [16] proposed a software-defined integrated space–ground network, which provided integrated communication services for various space missions. Reference [17] proposed a satellite–ground network composed of a ground network and a space network to provide users with global network coverage and less delay. Reference [18] proposed cognitive radio, which allowed satellite and ground networks to dynamically share their spectrum, so that the network obtained greater throughput. Reference [19] studied the connection delay between satellite networks in different long-distance intercontinental data communication scenarios, and proved the availability of satellite networks. It can be seen that the efficient transmission of large amounts of data and the limits of network coverage can now be addressed by the integrated satellite–ground network.

Resource allocation is an important research direction to improve the transmission performance of the satellite-–ground network [20]. Paper [21] established a joint communication and computation optimization model for satellite networks, which utilized stochastic geometry and queuing theory to achieve the best response delay. Reference [22] designed a dynamic channel resource allocation algorithm for the satellite ground integration architecture, which improved the number of Internet of Things terminal connections. The literature in [23] proposed a joint beamforming and carrier allocation scheme to solve the problem of spectrum shortage, which realized reliable satellite ground resource allocation. Reference [24] studied the problem of maximizing the capacity of all mobile terminals in an air–ground integrated network, and mainly used convex optimization theory to solve the problem. Reference [25] introduced some challenges faced by satellite–ground communication networks and proposed solutions based on artificial intelligence. The mentioned paper only wanted to improve the date rate of resources and the throughput of the system, and did not allocate resources from the needs of various entities in the network.

Matching is widely used in resource allocation [26]. The method of using matching theory to obtain optimal resource allocation in network resource allocation has become widespread. Two-sided matching has been widely studied in network resource allocation. Reference [27] designed an algorithm based on one-to-many matching, which transformed the network optimization problem into a one-to-many matching problem between spectrum resources and end users. Reference [28] proposed a matching-based theoretical framework to solve the issue of distributing computing resources in a three-layer fog computing network. The above papers only consider the case of two types of entities. However, in the satellite–ground network resource allocation, three different categories of entities must be taken into account equally. In social and commercial domains, three-sided matching problems are particularly prevalent [29,30]. Reference [31] utilized a three-sided matching algorithm to find the optimal matching between space–air–ground network facilities, content sources, and users. Paper [32] studied the problem of three-sided matching between spectrum, equipment, and user in wireless network virtualization. The above three-sided allocation method provides guidance for our study. However, as far as the author knows, the three-sided matching game is used for the first time in this research to address the issue of multi-party entity matching in the satellite–ground network.

## 3. System Model

In Figure 1, a cooperative satellite–ground network made up of GEO satellites, LEO satellites, IOT devices such as UAVs, PCs, and cars, as well as mobile users such as people living in remote places, is depicted. The user terminal uploads data to the LEO satellite, which then transmits it to the GEO satellite. During satellite–ground communication, the satellite directly establishes a satellite–ground link with the user terminal.

### 3.1. Network Model

A satellite–ground network scenario with users, GEO satellites, and LEO satellites is shown in Figure 1. The users are directly connected to the LEO satellite, which is directly connected to the GEO satellite. As the satellite moves frequently, it may eventually lose contact with the user. The duration of the GEO–LEO satellite link is measured in time slot t, with τ being the length of a time slot. In each time slot, since the duration of the U2L and L2G connections are both longer than τ, the topology of the satellite–ground network is essentially considered quasi-static. Use $U^{t}=\{u_{1}^{t},u_{2}^{t},\ldots ,u_{i}^{t},\ldots ,u_{N}^{t}\}$ to represent the user set in time slot *t*, $\forall i\in \left\{1,2,\ldots ,N\right\}$. The LEO satellite set is represented by $L^{t}=\{l_{1}^{t},l_{2}^{t},\ldots ,l_{j}^{t},\ldots ,l_{M}^{t}\}$, $\forall j\in \left\{1,2,\ldots M\right\}$. The GEO satellite set is represented by $G^{t}=\{g_{1}^{t},g_{2}^{t},\ldots ,g_{k}^{t},\ldots ,g_{K}^{t}\}$, $\forall k\in \left\{1,2,\ldots K\right\}$.

### 3.2. Delay Model

Since users are mainly located in remote areas, the connection between users and LEO satellites is obtained. The communication channel between the user and the satellite is mainly related to the distance from the user to the LEO satellite. According to [15], the received power of satellite $l_{j}^{t}$ at slot t  is:(1)$$P_{j,t}^{re}=\frac{P_{i}^{tr}G_{i}^{tr}G_{j}^{re}c}{4\pi fH_{i,j}^{t}}$$
where $G_{i}^{tr}$ represents the user’s transmit antenna gain, and $G_{j}^{re}$ represents the satellite’s receive antenna gain. $H_{i,j}^{t}$ is the distance from the LEO satellite to the user at slot t. c and f represent the speed of light and the carrier frequency, respectively. The achievable rate from user to satellite is given by:(2)$$R_{i,j,t}^{u\to s}=B_{i,j}^{t}log\left(1+\frac{P_{j,t}^{re}}{k_{B}T_{s}B_{i,j}^{t}}\right)$$
where $k_{B}$ is the Boltzmann constant and $T_{s}$ is the system noise temperature.

Assuming that all tasks are transferred to satellites via U2L channels, the transmission delay experienced by users in relation to LEO satellites may be calculated by dividing the task data size $\alpha_{i}^{t}$ by the associated transmission data rate $R_{i,j,t}^{u\to s}$
(3)$$T_{i,j,t}^{ul}=\frac{\alpha_{i}^{t}}{R_{i,j,t}^{u\to s}}$$

### 3.3. The Profitability of Satellite Model

This subsection describes and calculates the profitability of satellites. Since the primary purpose of commercial satellites is revenue generation, the profitability of the satellite constellation is also a major consideration.

The binary variable $\varphi_{i,j,t}$ represents whether the user $u_{i}^{t}$ is connected to the LEO satellite $l_{j}^{t}$ at slot t
(4)$$\varphi_{i,j,t}=\left\{\begin{array}{c} 1,\;\\ 0, \end{array}\right.\begin{array}{c} if\;user\;u_{i}^{t}\;is\;connected\;to\;LEO\;satellite\;l_{j}^{t}\;in\;slot\;t\;\;\\ otherwise \end{array}\;$$

The binary variable $\omega_{j,k,t}$ represents whether the LEO satellite $l_{j}^{t}$ is connected to the GEO satellite $g_{k}^{t}$ in slot t.
(5)$$\omega_{j,k,t}=\left\{\begin{array}{c} 1,\;\\ 0, \end{array}\right.\begin{array}{c} if\;\;LEO\;satellite\;l_{j}^{t}\;is\;connected\;to\;GEO\;satellite\;g_{k}^{t}\;in\;slot\;t\;\\ \mathrm{otherwise} \end{array}$$

We define the satellite’s total revenue $RS_{total}$ as the sum of the matching user-provided revenue as:(6)$$RS_{total}=\overset{N}{\underset{i=1}{\sum }}\overset{M}{\underset{j=1}{\sum }}\overset{K}{\underset{k=1}{\sum }}\overset{T}{\underset{t=1}{\sum }}\varphi_{i,j,t}\omega_{i,j,t}\varepsilon_{i,t}$$
where $\varepsilon_{i,t}$ is the user’s priority. Since high-priority users are likely to provide more important data and pay more for satellites, satellites are more willing to serve such users.

## 4. Problem Formulation

In **P0**, the goal is to maximize the total revenue of satellites. Since users with high priority need fewer delays and pay more for satellites, satellites are more willing to serve such users.
(7)$$P0:\;max\;RS_{total}$$
(8)$$s.t.\;\sum_{i=1}^{N}\varphi_{i,j,t}\leq N_{L}$$
(9)$$\overset{M}{\underset{j=1}{\sum }}\varphi_{i,j,t}\leq 1\;$$
(10)$$\overset{K}{\underset{k=1}{\sum }}\omega_{j,k,t}\leq N_{G}\;$$
(11)$$\varphi_{i,j,t}R_{i,j,t}\leq \delta_{i,j}\;$$
(12)$$\varphi_{i,j,t}\in \left\{0,1\right\}$$
(13)$$\omega_{j,k,t}\in \left\{0,1\right\}$$

Constraint (8) is the capacity constraint of LEO satellite $l_{j}^{t}$ and the maximum number of users matching each LEO satellite $l_{j}^{t}$ cannot exceed $N_{L}$ in each slot. Constraint (9) represents $u_{i}^{t}$, which can match with at most one LEO satellite in time slot t. Constraint (10) is the capacity constraint of $g_{k}^{t}$, and the maximum number of LEO satellites matching each GEO satellite cannot exceed $N_{G}$. Constraint (11) means the data transmission rate between $l_{j}^{t}$ and $u_{i}^{t}$ is limited by the channel capacity of $\;\delta_{i,j}$. Constraints (12) and (13) represent binary variables of $\varphi_{i,j,t}$ and $\omega_{j,k,t}$.

## 5. System Analysis

In this part, the resource allocation problem is solved using three-sided matching. Finding the most stable match with the highest cardinality in our suggested model is an NP-complete task. By creating some appropriate constraints, we convert it into an R-TMSC problem. We propose a three-sided cyclic matching algorithm for a GEO satellite.

### 5.1. Principle of Three-Sided Matching

Three-sided matching originates from two-sided matching. Two-sided matching is a matching method proposed by Gale and Shapely, and is widely used in marriage, person post, and other matching. With the increase in the types of matching subjects, the matching problem gradually expands [33]. Finding a stable outcome for each agent is the aim of the three-sided matching method [34]. Under the intuitive inspiration of the matching game framework, the relationship between user, LEO satellite, and GEO satellite can be described as a three-sided matching with size and cyclic preference list (TMSC). In TMSC, each type of participant has a preference list, which only includes one type of another agent, that is, GEO satellites only rank users, users only rank LEO satellites, and LEO satellites only rank GEO satellites. TMSC allows each agent to have multiple partners sorted by preference [35]. Let $\Delta^{t}=U^{t}\times L^{t}\times G^{t}$ represent the set of all possible triples. Therefore, $\Sigma^{t}\in \Delta^{t}$ is a subset. Definition 1 gives the blocking triples representing TMSC stability.

**Definition** **1.**
***Blocking Triple**: A triple ($u_{i}^{t}$, $l_{j}^{t}$, $g_{k}^{t}$)$\in \Delta^{t}$, but ($u_{i}^{t}$, $l_{j}^{t}$, $g_{k}^{t}$)$\notin \Sigma^{t}$, If all three members of $u_{i}^{t}$, $l_{j}^{t}$, and $g_{k}^{t}$ prefer triple ($u_{i}^{t}$, $l_{j}^{t}$, $g_{k}^{t}$) to at least one of their existing matched partners, then the triple is blocking.*


The aforementioned definition shows that each agent in the blocking triple is more eager to match each other as a triple than the current partners in matching $\Sigma^{t}$. The matching is deemed steady if $\Sigma^{t}$ lacks a blocking triple. The TMSC challenge in this study, which is inspired by [14], is to look for a stable matching with the highest cardinality $\left|\Sigma^{t}=\{(u_{i}^{t},\;l_{j}^{t},\;g_{k}^{t})\;\}\right|$.
(14)$$P1:\;\max\;\left|\Sigma^{t}\right|$$
(15)$$s.t.\;N\left(\Sigma^{t},u_{i}^{t}\right)\leq 1,\forall u_{i}^{t}\in U^{t}$$
(16)$$N\left(\Sigma^{t},g_{k}^{t}\right)\leq N_{G},\;\forall \;g_{k}^{t}\in G^{t}$$
(17)$$N\left(\Sigma^{t},\;l_{j}^{t}\right)\leq N_{L},\forall l_{j}^{t}\in L^{t}$$
(18)$$BT(u_{i}^{t},\;l_{j}^{t},\;g_{k}^{t})=0\;$$
where, $\left|\Sigma^{t}\right|$ is the number of triples. Constraint (15) states that a user can communicate with only one LEO satellite each time. Constraint (16) indicates the capacity limit $N_{G}$ of the GEO satellite, constraint (17) shows that a LEO satellite can only connect to, at most, $N_{L}$ GEO satellites, and constraint (18) ensures no blocking triple in $\Sigma^{t}$.

### 5.2. Three-Sided Cyclic Matching Algorithm

We added two restrictions from TMSC to form R-TMSC. Every GEO satellite has a user preference list that is derived from a master list. It is arranged in descending order based on the priority of the user. Users who demand lower delay will pay more and enjoy greater favor with GEO satellites. The preference list of all GEO satellites comes from the main list. In our example, the identical preference list $PL_{k}^{g}$ is generated by all GEO satellites. (We assume that every GEO satellite can serve every user.)
(19)$$PL_{k,i,t}^{g}=\varepsilon_{i,t}$$

On the other hand, according to the service quality measured by delay $T_{i,j,t}^{ul}$ (generate an acceptable set by Formula (3)), this ranks acceptable LEO satellites. Therefore, the user selects the LEO satellite according to the reciprocal of the expected delay. We represent the user’s preference list $PL_{i}^{u}$ as:(20)$$PL_{i,j,t}^{u}=1/T_{i,j,t}^{ul}$$

According to the previous subsection, the GEO satellite is independent of the LEO satellite. That is to say, each LEO satellite’s preference list $PL_{j}^{l}$ contains a constraint, and the ranking of all LEO satellites is the same, which can be expressed as:(21)$$PL_{j,k,t}^{l}=1$$

Before proposing our algorithm, the following notations are defined.
(22)$$\Pi^{+1}\left(\Sigma^{t},\;g_{k}^{t}\right)=\left\{\left.u_{i}^{t}\right|u_{i}^{t}{≻}_{\;g_{k}^{t}}\Sigma^{t}\left(g_{k}^{t}\right),u_{i}^{t}\in PL_{k,t}^{g}\right\}$$
represents the set of users that $g_{k}^{t}$ prefers over its current matching partner $\Sigma^{t}\left(\;g_{k}^{t}\right).$
(23)$$\;\Pi^{+1}\left(\Sigma^{t},\;u_{i}^{t}\right)=\left\{\left.l_{j}^{t}\right|l_{j}^{t}{≻}_{\;u_{i}^{t}}\Sigma^{t}\left(u_{i}^{t}\right),l_{j}^{t}\in PL_{i,t}^{u}\right\}$$
means the LEO satellite set that $u_{i}^{t}$ prefers to its present matching partner $\Sigma^{t}\left(u_{i}\right)$.
(24)$$\Pi^{-1}\left(\Sigma^{t},\;g_{k}^{t}\right)=\left\{\left.l_{j}^{t}\right|l_{j}^{t}\in L_{t},g_{k}^{t}\in PL_{j,t}^{g},\;N\left(\Sigma^{t},\;l_{j}^{t}\right)<N_{L}\right\}$$
represents the LEO satellite set that still has capacity for GEO satellite $g_{k}^{t}$.
(25)$$\Pi^{-2}\left(\Sigma^{t},\;g_{k}^{t}\right)=\left\{\left.u_{i}^{t}\right|\;\;\Pi^{+1}\left(\Sigma^{t},\;u_{i}^{t}\right)\cap \Pi^{-1}\left(\Sigma^{t},\;g_{k}^{t}\right)\neq \emptyset ,u_{i}^{t}\in U^{t}\right\}\;$$
indicates the user set, where a LEO satellite $l_{j}^{t}$ is preferred by $u_{i}^{t}$ above its current partner $\Sigma^{t}\left(u_{i}^{t}\right)$, and $l_{j}^{t}$ is also capable of matching with $g_{k}^{t}$.

According to above definitions, we propose Algorithm 1. Our algorithm is to look for the “best” triple and add it to the matching $\Sigma^{t}$ each time, starting from an empty set. Each “best” triple $(u_{i}^{t}$, $l_{j}^{t}$, $g_{k}^{t})$ is produced by first choosing a GEO satellite that meets specific criteria, after which this chosen GEO satellite selects the best user that satisfies its criteria, and finally this chosen user selects the most qualified LEO satellite.
**Algorithm 1**: Three-sided Matching Satellite–Ground Network Resource Allocation Algorithm 1. **Phase 1:** Initialization;2. **Input:** $U^{t},L^{t},G^{t}$;3. Set $\Sigma^{t}=\emptyset$, symbol=1;4. Phase 2: Matching;5. while symbol==1 perform;6.  Set symbol=0;7.    for each $g_{k}^{t}\in G^{t}$ perform;8.      $\overset{´}{\mu }=\Pi^{+1}(\Sigma^{t},\;g_{k}^{t})\cap \Pi^{-2}(\Sigma^{t},\;g_{k}^{t})$;9.          if $\overset{´}{\overset{´}{\mu }}\neq \emptyset$ then;10.            $u_{i}^{t}=Top(\overset{´}{\mu },g_{k}^{t}\;)$;11.            $\overset{´}{L}=\Pi^{+1}(\Sigma^{t},\;u_{i}^{t})\cap \;\Pi^{-1}(\Sigma^{t},\;g_{k}^{t})$;12.            $\;l_{j}^{t}=Top(\overset{´}{L},u_{i}^{t}\;)$;13.            if $N(\Sigma^{t},\;g_{k}^{t})==N_{G}$ then;14.            Choose the worst matched triple $\;{𝓉}_{worst}=(\Sigma^{t}(g_{k}^{t}),{\;\Sigma }^{t}(\Sigma^{t}(g_{k}^{t})),g_{k}^{t}\;)$ for $g_{k}^{t}$;15.              $\Sigma^{t}=\Sigma^{t}\backslash \;{𝓉}_{worst}$;16.              symbol=1;17.            end if:18.            if $N(\Sigma^{t},\;u_{i}^{t})==1$ then;19.              $\Sigma^{t}=\Sigma^{t}\backslash \;(u_{i}^{t},\Sigma^{t}(u_{i}^{t}),\;*)$;20.              symbol=1;21.            end if:22.                $\Sigma^{t}=\Sigma^{t}\cup (u_{i}^{t},l_{j}^{t},\;g_{k}^{t})$;23.            end if:24.    end for;25. end while:26. Output the stable result $\Sigma^{t}$.

Starting with an empty set, Algorithm 1 runs. As indicated in line 8, $\overset{´}{\mu }=\Pi^{+1}\left(\Sigma^{t},\;g_{k}^{t}\right)\cap \Pi^{-2}\left(\Sigma^{t},\;g_{k}^{t}\right)$ adds the better triples to $\Sigma^{t}$. If $\overset{´}{\overset{´}{\mu }}\neq \emptyset$ is true, line 10–23 will be executed. It is performed by choosing a more suitable user $u_{i}^{t}$ for $g_{k}^{t}$ in line 10. Note that $Top(\overset{´}{\mu },g_{k}^{t}\;)$ represents the user in $\overset{´}{\mu }$ that has the highest priority, and then performing the same for $u_{i}^{t}$ in line 13, choosing a more suitable LEO satellite. Finally, in line 22, this triple is added to $\Sigma^{t}$.

### 5.3. Performance Analysis

**Theorem 1.** *After a limited number of iterations, the algorithm will terminate and produce a stable match*.

**Proof of Theorem 1.** The algorithm will finish after a set number of steps because there are only so many LEO satellites and users on each user’s preference list. To prove that output matching is stable, we suppose that the output set $\Sigma^{t}$ is unstable. It implies that there must be a blocking triple ($u_{i}^{t}$, $l_{j}^{t}$, $g_{k}^{t}$), following that: $g_{k}^{t}\in PL_{j,t}^{l}$, $N(\Sigma^{t},\;l_{j}^{t})<N_{L},\;l_{j}^{t}{≻}_{\;u_{i}^{t}}\Sigma^{t}(u_{i}^{t})$, and $u_{i}^{t}{≻}_{\;g_{k}^{t}}\Sigma^{t}(g_{k}^{t})$. Thus, $u_{i}^{t}\in \Pi^{+1}(\Sigma^{t},\;g_{k}^{t}\;),\;\;l_{j}^{t}\in \Pi^{+1}(\Sigma^{t},\;u_{i}^{t}),\;\;l_{j}^{t}\in \Pi^{-1}(\Sigma^{t},\;g_{k}^{t}\;)$, and $u_{i}^{t}\in \Pi^{-2}(\Sigma^{t},\;g_{k}^{t})$. Then, $\Pi^{+1}(\Sigma^{t},\;g_{k}^{t}\;)\cap \Pi^{-2}(\Sigma^{t},\;g_{k}^{t}\;)\neq \emptyset$, and $\Pi^{+1}(\Sigma^{t},\;u_{i}^{t})\cap \Pi^{-1}(\Sigma^{t},\;g_{k}^{t}\;)\neq \emptyset$. In this situation, the algorithm will not terminate, which is paradoxical. Therefore, the output result $\Sigma^{t}$ must be stable [12]. □ 

**Theorem 2.** 
*In O($KTM\sum_{u_{i\in U}}|PL_{i,j,t}^{u}|$) iterations, the algorithm can always identify a stable match.*


**Proof of Theorem 2.** If a LEO satellite has a large capacity $N_{L}$, only one user will be assigned to that LEO satellite during each iteration. Thus, the total time T, user’ number N, LEO satellite’ number M, and GEO satellite’ number K determine the maximum amount of time needed, hence, this algorithm’s time complexity is $O\left(KTMN\right)$. However, when $N_{L}$ is very limited, just one user is assigned to a more suitable LEO satellite in its preference list during each sign iteration, continuing until the sign equals 0 and this algorithm ends. In reality, each user pre-matches every LEO satellite in its preference list when the algorithm is running, which is the worst scenario. Therefore, the length of the user’s preference list, rather than the matching set $\Sigma^{t}$, dictates the maximum time required, so the time complexity is O($KTM\sum_{u_{i}^{t}\in U^{t}}|PL_{i,j,t}^{u}|$). □

## 6. Performance Results and Analysis

In this part, we choose to compare with two-sided matching method, greedy matching solution, and random matching solution, and evaluate the performance of the proposed algorithm through MATLAB simulation. The algorithms used in the simulation are as follows:(1)Proposed method: a three-sided cycle matching method;(2)Two-sided matching method: it decouples three types of entities into two types for matching;(3)Greedy matching method: it finds the local optimal solution by using the best option in the current situation;(4)Random matching method: random matching of three kinds of entities

The main simulation parameters are listed in Table 1 [15,36]. Within a geographical area of 10 km × 10 km, the users are dispersed at random. There are six LEO satellites, six GEO satellites, and $N\in \left[20,\;140\right]$ users in total. Specifically, the orbit height of the LEO satellite is 780 km, the orbit period is 6028 s, and the LEO satellite’s capacity $N_{L}$ is 20, and the GEO satellite’s capacity $N_{G}$ is 6.

Figure 2 compares the four methods from the perspective of average delay.

In the delay model, the transmission delay experienced by users connected to LEO satellites may be calculated by dividing the task data size $\alpha_{i}^{t}$ by the associated transmission data rate $R_{i,j,t}^{u\to s}$. As can be seen in Figure 2, with the increase in users, the average delay of users is also increasing. Our proposed algorithm can always produce a stable match so it has the lowest user delay. The delay of the two-sided matching and greedy algorithm is higher than the proposed three-sided cyclic matching algorithm, and the random allocation is the worst.

In Figure 3, the number of users served by the four algorithms is depicted as the user base grows. We increase the number of users from 20 to 140 in 20 steps. According to the figure, we can see that the proposed algorithm, two-sided matching algorithm, and greedy method serve all users until the user number reaches 100. When there are about 120 users, the two-sided matching algorithm tends to be flat. Since then, when the number of users continues to rise, the proposed algorithm tends to be stable when serving about 120 users, because the available LEO satellite capacity is limited. It can be seen that our algorithm is superior to other algorithms in the number of service users.

Figure 4 and Figure 5 compare the change in satellite revenue with users and the change in satellite revenue at each slot. We increase the number of users from 20 to 140 in 20 steps. As shown in Figure 4, under all four scenarios, as more users join the network, the user’s income increases. This is because the more users of the service, the more revenue. We can see that when the number of users reaches 120, the revenue is still increasing, because new users may have higher priority. On the other hand, Figure 5 shows that the revenue of satellites increases as time slots increase. The more users successfully matched, the more revenue the satellite obtains, so it shows an increasing trend. We can also observe from Figure 4 and Figure 5 that our proposed algorithm is superior to two-sided matching, greedy, and random allocation methods.

We further study the impact of time slot changes on the number of served users. For clarity, we considered the case where the user set is 120. In addition, we need to assume that the user’s requirements for delay do not change in each time slot, so that we can make a fair comparison. As shown in Figure 6, the algorithm proposed by us has an obvious growth trend, and it serves more users than the two-sided matching. The greedy algorithm and the random algorithm are far lower than the three-sided matching algorithm. The number of users served by these four algorithms increases with the increase in time slots.

## 7. Conclusions

In this paper, we propose a resource allocation model for the satellite–ground network, where the relationship between GEO satellite, LEO satellite, and user is established. GEO satellites provide data backhaul for user, and LEO satellites provide data transmission services according to users’ requests. Therefore, we propose the three-sided cyclic matching algorithm for GEO satellites to solve the above problems. The simulation results verify that the proposed three-sided cyclic matching algorithm is superior to the traditional algorithm in terms of time delay and satellite revenue.

## Figures and Tables

**Figure 1 sensors-22-08436-f001:**
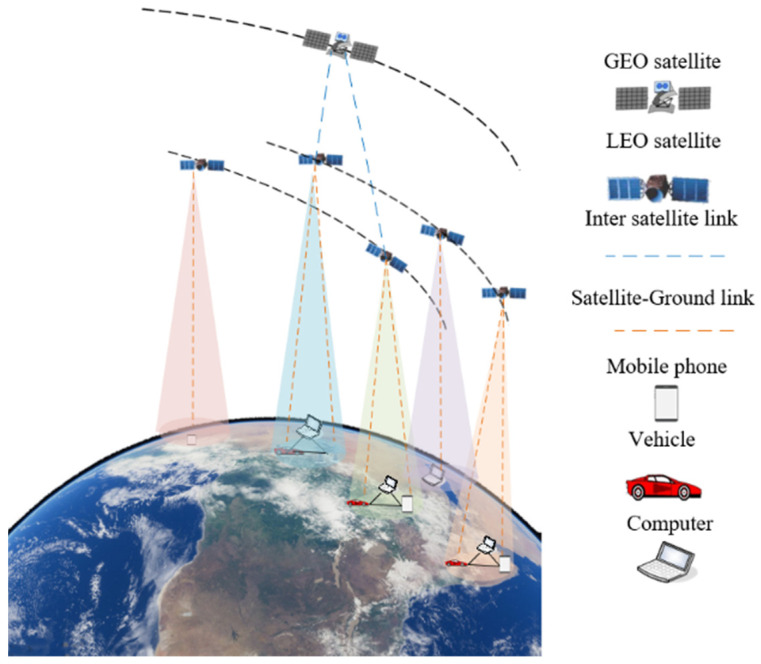
System model.

**Figure 2 sensors-22-08436-f002:**
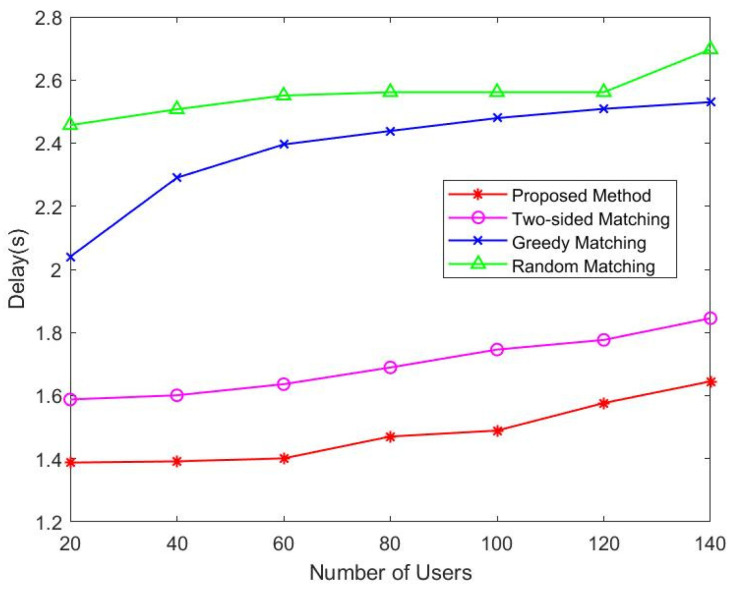
Average delay vs. users.

**Figure 3 sensors-22-08436-f003:**
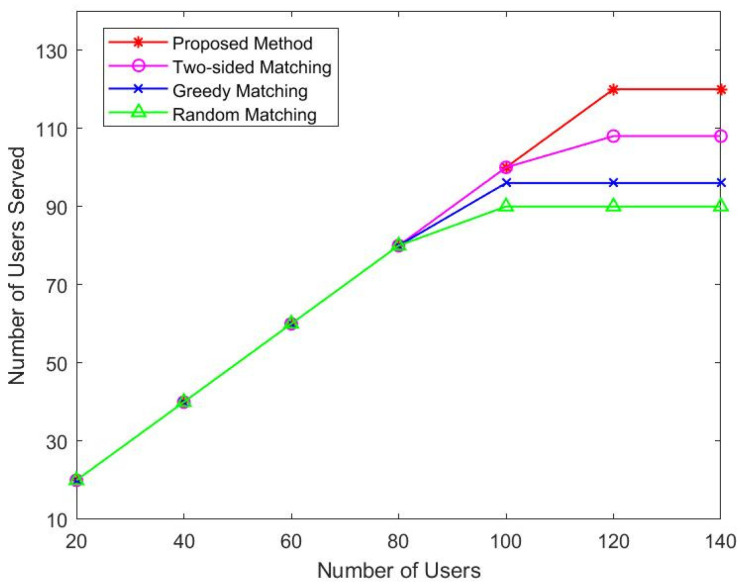
Number of output matching vs. users.

**Figure 4 sensors-22-08436-f004:**
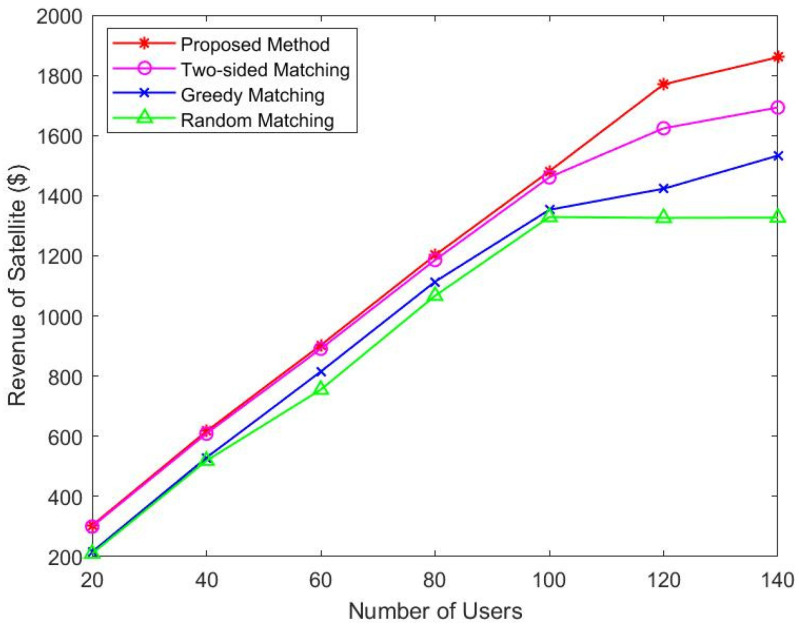
Revenue of satellites vs. users.

**Figure 5 sensors-22-08436-f005:**
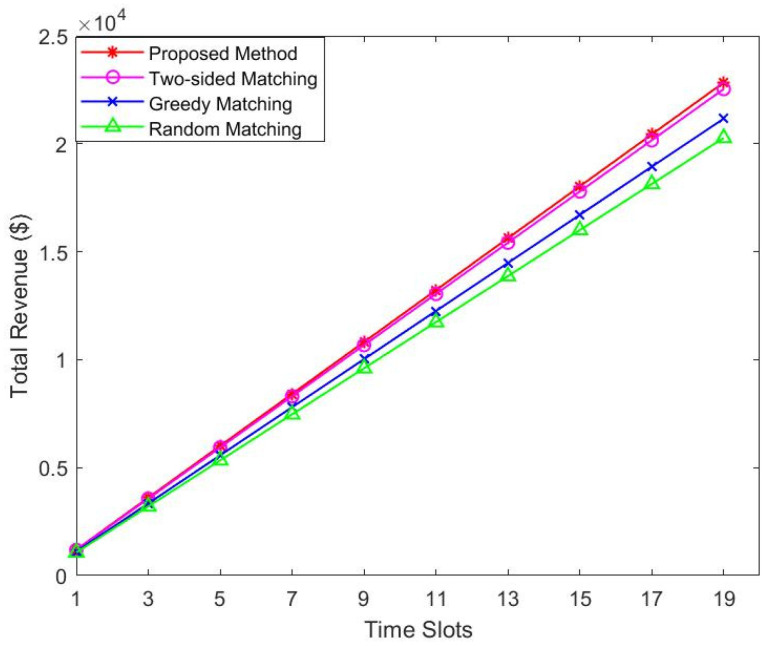
Total revenue of satellites vs. time slots.

**Figure 6 sensors-22-08436-f006:**
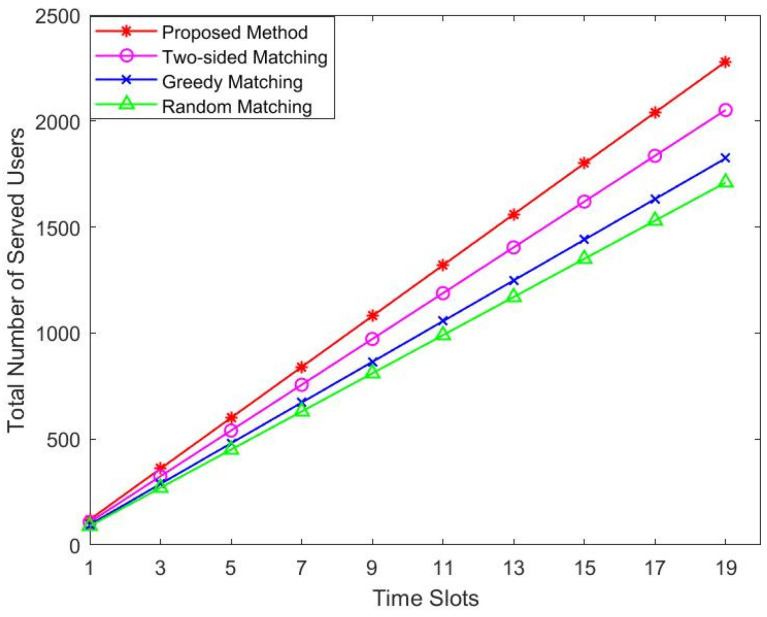
Total number of served users vs. time slots.

**Table 1 sensors-22-08436-t001:** Simulation parameters.

Parameter	Value	Parameter	Value
$B_{i,j}^{t}$	54 MHz	f	2.4 GHz
$G_{i}^{tr}G_{j}^{re}$	15 dB	$P_{i}^{tr}$	0.5 W
$\alpha_{i}^{t}$	[10, 100] Mbits	c	3 × 10^8^ m/s
$k_{B}$	1.38 × 10^−23^ J/K	$T_{s}$	1000 J
$N_{L}$	6	$N_{G}$	20
M	6	K	2
N	[20, 140]		

## Data Availability

Not applicable.

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
