# Peer review of "Matching-Based Resource Allocation for Satellite–Ground Network"

_sensors, 2022, doi:10.3390/s22218436_

Round 1

Reviewer 1 Report

This paper proposed the three-sided cyclic matching algorithm for a resource allocation model for satellite-ground network, where the relationship between GEO satellite, LEO satellite and user is established. The simulation results verified that the proposed algorithm is better than the traditional algorithm in terms of time delay and satellite revenue.

There are several aspects that the paper could be improved.

1.      The main contributions of this paper only described what the author did, but the motivation and the novelty of the work is not so clear. The authors may demonstrate more about the difference compared with other existing research, especially with two-sided cyclic matching.

2.      Authors listed the proposed algorithm, but the theoretical analysis is needed to illustrate how the algorithm come into being.

3.      There may be some doubts about fig. 3 and fig.4. Since the number of users served is already limited at 120 as in fig.3, how can the revenue still increase after that as in fig.4? Also, the results from fig. 3 and fig.4 seemed to be apparent,the number of users served and the revenue surely will increased by the numbers of users. The authors could add more experiments or simulations from different angels.

4.      The editing needs to be improved. Several errors in the paper. For example, line 178~179, Constraint (14)~(15) should be 11~12.

Reviewer 2 Report

In this manuscript the authors propose the three-sided cyclic matching algorithm for GEO satellites to solve the above problems.The results reflect the efficiency of the simulated algorithm. This is a excellent work.

In this manuscript there are some grammatical errors and the inappropriate use of the comma. So I suggest that the authors do an English review.

Reviewer 3 Report

Introduction section needs more background information about the topic

All figures need more explanation in the text.

Related work section needs more work. More articles should be explained in this section.

Why the authors choose two-sided matching method, greedy matching solution and random matching solution for comparison? Are these the best methods available? Not clear

How the delay will change if number of users are more than 140?

More results are needed to gauge the effectiveness of proposed method.

English of the manuscript needs to be improved

Round 2

Reviewer 1 Report

The review comments were answered in detail and well, and the quality of the article was improved to meet the publishing requirements.

Reviewer 3 Report

The reviewer comments are answered in detail and the quality of the article is improved to meet the publishing requirements.